# Severe Impairment of Left Ventricular Regional Strain in STEMI Patients Is Associated with Post-Infarct Remodeling

**DOI:** 10.3390/jcm11185348

**Published:** 2022-09-12

**Authors:** Giovanni Andrea Luisi, Gabriele Pestelli, Giulia Lorenzoni, Filippo Trevisan, Vittorio Smarrazzo, Andrea Fiorencis, Filippo Flamigni, Roberto Ferrari, Donato Mele

**Affiliations:** 1Cardiology Unit, University Hospital of Ferrara, 44124 Ferrara, Italy; 2Department of Cardiac, Thoracic, Vascular Sciences and Public Health, University of Padova, 35128 Padova, Italy

**Keywords:** myocardial infarction, speckle tracking echocardiography, global longitudinal strain, left ventricular remodeling, severely altered strain

## Abstract

Background: Measures of global left ventricular (LV) systolic function have limitations for the prediction of post-infarct LV remodeling (LVR). Therefore, we tested the association between a new measure of regional LV systolic function—the percentage of severely altered strain (%SAS)- and LVR after acute ST-elevation myocardial infarction (STEMI). As a secondary objective, we also evaluated the association between %SAS and clinical events during follow-up. Methods: Of 177 patients undergoing echocardiography within 24 h from primary percutaneous coronary angioplasty, 172 were studied for 3 months, 167 for 12 months, and 10 died. The %SAS was calculated by dividing the number of LV myocardial segments with ≥−5% peak systolic longitudinal strain by the total number of segments. LVR was defined as the increase in end-diastolic volume >20% at its first occurrence compared to baseline. Results: LVR percentage was 10.2% and 15.8% at 3 and 12 months, respectively. Based on univariable analysis, a number of clinical, laboratory, electrocardiographic and echocardiographic variables were associated with LVR. Based on multivariable analysis, %SAS and TnI peak remained associated with LVR (for %SAS 5% increase, OR 1.226, 95% CI 1.098–1.369, *p* < 0.0005; for TnI peak, OR 1.025, 95% CI 1.004–1.047, *p* = 0.022). %SAS and LVR were also associated with occurrence of clinical events at a median follow-up of 43 months (HR 1.02, 95% CI 1.0–1.04, *p* = 0.0165). Conclusions: In patients treated for acute STEMI, acute %SAS is associated with post-infarct LVR. Therefore, we suggest performing such evaluations on a routine basis to identify, as early as possible, STEMI patients at higher risk.

## 1. Introduction

Left ventricular remodeling (LVR) is a complication of acute myocardial infarction (MI). It has a negative prognostic value and is the main cause of heart failure in acute MI survivors [1,2]. Several studies sought to identify the predictors of LVR after acute MI. Initial echocardiographic investigations recognized LV ejection fraction (EF), end-systolic volume (ESV), and wall motion score index (WMSI) as LVR predictors [3,4,5]. In recent years, additional predictors based on advanced echocardiographic techniques, such as speckle tracking echocardiography (STE), have emerged, particularly global longitudinal strain (GLS) [6,7]. All of these echocardiographic measures, however, are limited, because they represent global, not regional LV function and are influenced not only by areas of depressed myocardial function (scarred or stunned) but also by remote normal or hypercontractile myocardium. 

In this study, we sought to verify whether a new measure of regional longitudinal strain—the percentage of severely altered strain (%SAS)—evaluated by STE during the acute phase of ST-elevation MI (STEMI) is associated with subsequent development of LVR and is better than the other echocardiographic predictors. The rationale underlying the use of %SAS is that it represents the amount of acute severely injured myocardium, without including mildly injured, normal or hypercontractile LV areas, thus avoiding a fundamental limitation of global LV function indices.

## 2. Materials and Methods

### 2.1. Study Patients 

Patients were referred to the catheterization laboratory of the Cardiac Unit of the University Hospital of Ferrara for primary PCI for suspected STEMI between September 2015 and 2017. The diagnosis of STEMI was made according to standard clinical and electrocardiographic criteria [8] and confirmed at the moment of coronary angiography. Inclusion criteria were: (1) age ≥ 18 years, (2) coronary artery disease treated with PCI, (3) two-dimensional (2D) echocardiography performed within 24 h from admission to the Intensive Care Unit (ICU) after PCI (T0 time), and (4) at least one 2D echocardiographic examination performed within 1 year from discharge. Exclusion criteria were: (1) inadequate echocardiographic image quality (defined as inability to evaluate all three standard apical views for strain analysis), (2) absence of sinus rhythm in echocardiography performed at T0, and, (3) death before hospital discharge.

### 2.2. Study Design and Objectives

This was a single-center observational study. According to the follow-up program for STEMI patients treated at the Ferrara Cardiac Unit, all patients were scheduled for echocardiography 3 months (T1 time) and 12 months (T2 time) after discharge.

The primary objective of the study was to verify whether acute %SAS (at T0) is associated with the development of LVR. Because LVR is a progressive phenomenon detected echocardiographically up to 1 year from the acute event [9], the 12-month period from patient discharge was considered for LVR identification. LVR was defined as an increase in end-diastolic volume (EDV) > 20% relatively to the baseline value at the moment of its first recognition [10], which could be at T1 or T2 from hospital discharge.

To provide information about the clinical value of acute %SAS, we also verified whether this measure was associated with clinical events during follow-up (secondary objective). For this purpose, death from any cause and re-hospitalization for heart failure or acute coronary syndrome were chosen as clinical events and grouped into a composite endpoint. Outcome status was assessed by review of the medical records through the hospital medical informatic platform. If a patient had more than one clinical event, the first one that occurred temporally was considered. The median duration of follow-up from STEMI occurrence was 43 months (interquartile range: 36–51 months).

### 2.3. Conventional and Speckle Tracking Echocardiography

(A) Image acquisition. The 2D echocardiographic examination was performed using commercial echo scanners (Vivid 7 and E9, GE Medical Systems, Milwaukee, WI, USA) equipped with a 3.5 MHz phased-array transducer. All Doppler and echocardiographic acquisitions were obtained at held end-expiration. Sector width was optimized to allow visualization of the whole LV myocardium in the 3 standard apical views (4-chamber, 2-chamber and long-axis) and to maximize frame rate (which varied between 60 and 100 fps). (B) Image analysis. Analysis of the echocardiographic images was performed off-line using a commercially available software (EchoPAC PC 112 rev 1.3, GE Medical Systems, Milwaukee, WI, USA). LV end-diastolic volume (EDV), ESV, EF and left atrial volume index (LAVI) were calculated using the biplane Simpson’s method [11]. LV diastolic function and the degree of mitral regurgitation (MR) were assessed according to specific guidelines [12,13]. WMSI was calculated using an LV 16-segment model, including six myocardial segments at the basal and mid-portion level and four segments at the apex [14]. The percentage of the extent of wall motion abnormalities (%WMA), was obtained by dividing the number of akinetic and dyskinetic segments by the total number of segments evaluated [15]. GLS was provided automatically by the software as the average value of the peak systolic longitudinal strain of each myocardial segment for the three standard apical views. For the evaluation of %SAS, myocardial segments with peak systolic longitudinal strain ≥−5% were considered severely dysfunctional myocardium [16]. The %SAS index was obtained by dividing the number of segments with longitudinal strain ≥−5% by the total number of segments evaluated. For both GLS and %SAS, an LV 18-segment model was used, with six myocardial segments at the basal, mid-portion and apical levels. The investigators who analyzed the images were blinded to the patients’ clinical information.

### 2.4. Statistical Analysis

After testing for normal distribution with the Kolmogorov–Smirnov test, continuous variables were expressed as median values with 25th and 75th percentiles. Categorical variables were expressed as percentages. Variables of patients with and without LVR or with and without clinical events were compared using the Mann–Whitney U test or the chi-square test, when appropriate. Values of continuous variables at T0, T1 and T2 were compared using the Friedman test, and pairwise comparisons were performed with a Bonferroni correction.

For the primary objective, a univariable logistic regression, including the clinical and echocardiographic variables, was performed to evaluate the association with LVR. For the multivariable analysis, the following variables were selected a priori: time from symptoms’ onset to STEMI diagnosis ≥ 3 h, peak of TnI, ST-resolution, use of MRAs at discharge, LV ESV, EF, GLS, WMSI, %WMA, and %SAS. Except for %SAS, all of these variables are known to be associated with LVR from previous studies [3,4,5,6,7,14,15,17,18,19]. Variables were tested for multicollinearity with %SAS using the variance inflation factor (VIF), with a value ≥ 5 indicating presence of significant collinearity. Different multivariable stepwise logistic regression analyses were performed, each one including only variables with VIF value < 5. The odds ratio (OR) for each statistically significant variable was calculated together with the 95% confidence interval (CI). Receiver-operating characteristic (ROC) curve analysis was used to calculate the area under the curve (AUC) and the 95% CI for the multivariable models.

For the secondary objective, we performed a univariable and a multivariable Cox analysis. The hazard ratio (HR) for each statistically significant variable was calculated together with the 95% CI. For the multivariable Cox analysis, it was decided a priori to include in the model LVR those variables significantly associated with LVR in the multivariable analysis for the primary objective.

Intra and interobserver variability for %SAS measurement were assessed in 20 randomly selected patients at T0 using the interclass correlation coefficient (ICC) and the Bland–Altman analysis, calculating bias and limits of agreement (LOAs). The statistical analysis was performed using the IBM SPSS Statistics software, v. 25 (IBM, Armonk, NY, USA). A *p* value < 0.05 was considered statistically significant.

## 3. Results

Five-hundred forty-eight patients referred to the catheterization laboratory with STEMI diagnosis confirmed by coronary angiography were initially considered. Then, 314 patients were excluded because they were transferred to other hospitals of the Ferrara STEMI network after PCI (*n* = 303) or died before hospital discharge (*n* = 11). Of the 234 remaining patients, another 57 were excluded: 8 had atrial fibrillation at T0 echocardiography, 22 had inadequate echocardiographic image quality, and 27 were lost to follow-up. Therefore, 177 patients constituted the final study cohort. Five of these patients died before echocardiography at T1 and 5 before echocardiography at T2.

The echocardiographic examination was performed at T0 within 1 ± 1 day from ICU admission in 177 patients, at T1 after 131 ± 51 days in 172 patients and at T2 after 469 ± 91 days in 167 patients. After the acute event, 30 (16.9%) patients had LVR; 18 (60%) of them had first evidence of LVR at T1 and 12 (40%) at T2. One patient with LVR at T1 died before T2, and another one showed reversed LVR at T2. The percentage of LVR was 10.2% (18/177) at T1 and 15.8% (28/177) at T2. Of the 28 patients with LVR at T2, 12 (43%) did not show LVR at T1.

### 3.1. Clinical Characteristics

The main clinical characteristics of patients are reported in Table 1. In the overall cohort, median age was 67 years, most patients were men (72.3%) and many had cardiovascular risk factors. The most common infarct site was the anterior one (44.9%), 28.8% of patients had a 3-vessel disease, and 12.3% a previous MI. Although the vast majority of patients had complete culprit coronary reperfusion (TIMI flow 3 in 96% of cases), 37.9% had no ST-resolution > 50% when ECG was performed in the ICU within 4 h from primary PCI, consistent with the “no-reflow” phenomenon. Medical therapy at discharge was optimized for all patients (Appendix A).

No significant differences were observed between patients with and without LVR in terms of age, gender, body surface area, body mass index, cardiovascular risk factors, systolic blood pressure, heart rate, site of infarction, culprit coronary artery, final TIMI flow, time from first medical contact to PCI and staged PCI during hospitalization (Table 1). LVR patients had higher levels of peak of troponin (Tn) I and creatine kinase-MB (CK-MB), more frequently Killip class 3–4, a longer time from symptoms’ onset to STEMI diagnosis, and a lower percentage of ST-resolution (Table 1). Additionally, LVR patients were treated more frequently with mineralocorticoid receptor antagonists (MRAs), both at discharge and at 12 months (Appendix A).

### 3.2. Echocardiographic Characteristics

Baseline, T1 and T2 echocardiographic characteristics are reported in Table 2 and Table 3. In the overall cohort, EDV did not change over time, whereas ESV decreased. Parameters of global and regional LV systolic function all improved during follow-up, while MR severity decreased. LAVI increased over time. 

In patients without LVR, EDV did not change and ESV progressively decreased over time. Global and regional measures of LV systolic function improved. %SAS reduced markedly. LAVI had a modest, although significant, increase. MR severity decreased at T1 and was maintained at T2. E/e’ ratio decreased progressively. 

Compared to non-LVR patients, at baseline patients with LVR did not differ in terms of EDV but had lower EF and GLS (in absolute values), and higher ESV, WMSI, %WMA and %SAS. LAVI, MR severity and diastolic measures did not differ. In patients with LVR, EDV and ESV progressively increased over time, and LV-EF improved only mildly. GLS improved, and WMSI did not vary. Regional measures of LV systolic function, such as %WMA and %SAS, declined during follow-up, especially at T2. LAVI increased at T2. MR severity remained similar over time and worse than that in patients without LVR at T1 and T2.

### 3.3. Primary Objective: Left Ventricular Remodeling

In the univariable analysis, time from symptoms’ onset to STEMI diagnosis ≥ 3 h, time from first medical contact to PCI, infarct site, ST-resolution, peak of TnI, peak of CK-MB, glycated hemoglobin, use of MRAs at discharge, LV ESV, EF, GLS, WMSI, %WMA, and %SAS were associated with LVR (Appendix A). A multicollinearity was found between some variables (WMSI and %WMA) and %SAS, thus three multivariable models were generated.

Model 1 excluded WMSI and %WMA (Table 4). In this model, only %SAS and peak of TnI remained associated with LVR: for %SAS 5% increase, the OR was 1.226 (95% CI, 1.098–1.369, *p* < 0.0005) while for peak of TnI, the OR was 1.025 (95% CI, 1.004–1.047, *p* = 0.022). The ROC curve analysis of this multivariable model showed an AUC of 0.82 (95% CI 0.73–0.91, *p* < 0.0005) (Figure 1A); AUC of %SAS and TnI were 0.80 (95% CI 0.71–0.88) and 0.76 (95% CI 0.66–0.86; *p* = 0.483 vs. %SAS), respectively. Optimal baseline cut-off values were 44% for %SAS (sensitivity 76.7%, specificity 77.2%) and 49 ng/mL for peak of TnI (sensitivity 82.8%, specificity 61%) (Figure 1B,C). Correlation between %SAS and peak of TnI was 0.45 (*p* < 0.0005). 

Model 2 and 3 excluded %SAS and included WMSI and %WMA separately (Table 4). Neither WMSI nor %WMA were associated with LVR. Of note, in these two models, GLS became significantly associated with LVR, whereas it was not in model 1 when %SAS was included in the analysis. LV-EF and ESV were not associated with LVR in all three models.

Figure 2 and Figure 3 show the example of two patients with similar LV-EF but very different %SAS. Only the patient with high %SAS developed LVR (Figure 2).

### 3.4. Secondary Objective: Clinical Events

During the median follow-up period of 43 months, 11 patients died, 15 were re-hospitalized for heart failure, and 15 for acute coronary syndrome as first clinical event, for a total of 41 first events in 41 patients (23% of the entire patient cohort). Because 11 patients had more than one clinical event, the overall clinical events were 55. Survival probability free of clinical events is shown in Figure 4.

Clinical and echocardiographic characteristics of patients with and without clinical events are reported in Appendix A. Patients of the two groups differed in age, cardiovascular risk factors and echocardiographic measures. Of note, while baseline %SAS markedly differed between the two groups (23% vs. 41%, *p* = 0.003), TnI at T0 was not different between patients with and without clinical events.

Baseline variables associated with clinical events in univariable Cox analysis are reported in Appendix A. LVR was associated with occurrence of clinical events during follow-up (HR 2.53, 95% CI 1.19–5.34, *p* = 0.015). In addition to LVR, multivariable Cox analysis included %SAS and peak of TnI (GLS was not included, since it was not significantly associated with LVR when %SAS and peak of TnI were considered (Table 4, model 1)). Only %SAS (HR 1.02, 95% CI 1.0–1.04, *p* = 0.0165) and LVR (HR 1.93, 95% CI 0.80–4.66, *p* = 0.0148) were associated with occurrence of clinical events during follow-up, while peak of TnI was not (HR 0.99, 95% CI 0.97–1.01, *p* = 0.2264).

### 3.5. Observer Variability for %SAS Calculation

As regards the intraobserver variability of %SAS calculation, the ICC was 0.993, bias −0.2%, and LOAs 5.4 and −5.9%; as regards the interobserver variability, the ICC was 0.975, bias −0.1%, and LOAs 8.7 and −9.9%.

## 4. Discussion

This study explored, in patients treated for STEMI, the association between acute echocardiographic measures of LV systolic function and LVR (primary endpoint) and clinical events during follow-up (secondary endpoint). The main findings were: (1) in patients treated for STEMI, acute %SAS was associated with LVR; (2) this association was not observed for conventional echocardiographic measures of global LV function in the multivariable analysis, while it was observed for peak of TnI; (3) acute %SAS was also associated with an unfavorable clinical outcome, while peak of TnI was not; (4) a significant proportion (43%) of patients with LVR at 12 months did not have LVR at 3 months; (5) LVR at its first occurrence was associated with clinical outcome. All of these findings have implications for the acute evaluation and follow-up strategy of patients who experience a STEMI.

### 4.1. Rate and Significance of Post-Infarct LVR

Investigations performed in the decade between 1996 and 2006 reported a rate of LVR of 16–31% between 6 and 12 months after revascularized acute STEMI [4,9,10]. More recently, a registry published in 2020 (collected since February 2004) [14] showed that 48% of 1995 patients with STEMI developed LVR in the first 12 months of follow-up and the majority (64%) during the first 3 months. Because primary PCI techniques and pharmacotherapy have evolved during the time frame of the registry, these factors could not be accounted for.

We observed a percentage of LVR of 10.2% at 3 months and 15.8% at 1 year. In our opinion, this lower LVR prevalence compared to previous studies could be related to the effectiveness of contemporary STEMI treatments (current PCI techniques, staged revascularization and updated pharmacotherapy). On the other hand, the persistence of a significant number of patients with LVR also evidences that development of post-infarct LVR is still affected by determinants not fully controlled by current therapeutic strategies (for example, the no-reflow phenomenon).

In our patients, LVR was significantly associated with an unfavorable clinical outcome in univariable and multivariable analysis. This confirms previous observations [1,14] and underlines the importance of evaluating LVR during the post-infarct follow-up.

### 4.2. Determinants of Post-Infarct LVR

Various echocardiographic measures of LV global systolic function, including LV-EF, ESV, WMSI and GLS, have been proposed over the years to predict, during the early phase of STEMI, subsequent development of LVR or outcome [3,4,5,6,15,16]. Results, however, were not univocal. For example, in a meta-analysis, 2D GLS, evaluated within 48 h after the acute event, has been shown to predict adverse LVR after STEMI (defined as an increase in LV-EDV and/or LV-ESV by 15 to 20%) [6]. Other authors, however, found that, in high-risk STEMI patients, GLS was not superior to conventional echocardiography in predicting outcome [20]. 

In our study, LV-EF, ESV, and WMSI were not associated with LVR. GLS was significantly associated with LVR only when %SAS was not included in the multivariable analysis (Table 4). A possible explanation for these findings is that the global measures of LV systolic function not only depend on irreversibly injured cardiac muscle but also on mildly injured, stunned, remote normal and hypercontractile myocardium, which may variably combine in individual patients. 

A different approach to prediction of LVR is the evaluation of regional LV systolic function. This approach has been shown to predict LVR in previous investigations based on conventional [21] and STE imaging [22] and seems also better than GLS [22]. In the present study, we suggested a new index to evaluate regional LV systolic function (%SAS), which relies on recognition of those LV segments with severely reduced myocardial contraction. We observed that the percentage of these segments is significantly and independently associated with LVR. 

Of note, in LVR patients, the baseline %SAS was high (54% on average). This could be related to the very early evaluation of %SAS in the acute phase of STEMI, when infarct size is still far from stabilization. Additionally, because a larger acute infarct size is associated with the presence of microvascular obstruction [23], the high %SAS observed at baseline very likely includes severe myocardial injury and, therefore, has the potential to predict LVR and outcome. 

In our patient cohort, peak of TnI was also significantly and independently associated with LVR. This finding agrees with that of other authors, showing that patients with LVR after STEMI were characterized by higher levels of peak TnI [14]. The association between peak of Tn and LVR can be explained considering that peak of Tn is also an expression of the infarct size [24], a determinant of LVR [25]. Because in our study the AUC of TnI and %SAS in predicting LVR were similar (0.76 vs. 0.80, *p* = 0.483; Figure 1), one could wonder what the advantage of measuring %SAS instead of TnI is. There are several answers to this question.

First, multiple Tn assays are needed after the STEMI diagnosis to identify the peak value of Tn, whereas %SAS is a single evaluation, and this may facilitate very early prediction of LVR. Second, the correlation between serum Tn after STEMI and infarct size is impaired in case of slow or incomplete reperfusion and renal insufficiency, which delay peak of Tn [26], while kidney disease does not influence %SAS. Third, while %SAS was different in our patients with and without clinical events, peak of TnI was not (Appendix A); also, TnI was not associated with clinical outcome in univariable and multivariable Cox analysis. This agrees with recent observations showing that the prognostic implications of the peak Tn value seem to be minimal in revascularized patients with acute coronary syndrome [27] and even absent in STEMI patients in the contemporary era [28]. Therefore, in comparison with peak TnI, %SAS assessment shortly after primary PCI in patients with acute STEMI could be more practical and informative.

### 4.3. Evolution of %SAS over Time

In our patient cohort, %SAS progressively decreased over time after the acute STEMI, with a parallel increase in GLS (Table 2). This is in agreement with progressive regression in MI size from the acute to the subacute and chronic phase, which is the consequence of the gradual resolution of myocardial edema, intramyocardial hemorrhage, and microvascular obstruction [23]. A contrast cardiac magnetic resonance study also showed that progressive reduction in late enhancement in the infarct region is associated with an increase in contractility in the same segments [29], thus linking regression in MI size with improvement in myocardial function. 

In our investigation, the %SAS decrease occurred both in patients with and without LVR, but in those with LVR, the magnitude of reduction was much less (Table 3). This can be related to both a larger infarct size and to severe microvascular obstruction. An indirect confirmation of the role of microvascular obstruction in the LVR subgroup comes from the higher percentage of patients without ST-resolution after primary PCI in this subgroup (Table 1). 

Finally, in LVR patients, the median %SAS at 12 months was 12%. This represents the residual infarct size, that is, the non-viable myocardium with transmural scar tissue [30].

### 4.4. Advantages of %SAS

The %SAS is an objective measure of regional LV systolic function. This is an advantage compared to other parameters, such as WMSI and %WMA, based on visual interpretation of myocardial contraction. Further, calculation of %SAS is simple and relatively fast with current echocardiographic technology. In fact, the number of segments with severe alteration of strain can be easily obtained just by looking at the LV polar plot automatically provided by the echo scanner (Figure 2 and Figure 3).

### 4.5. Study Limitations

This study was a single-center investigation. This implies a limited number of STEMI patients but had the advantage of a homogeneous approach to treatment and follow-up strategy. In general, patients transferred rapidly after primary PCI to the other hospitals of the STEMI network were clinically more stable or had less residual coronary disease needing deferred PCI. This may have introduced a bias at baseline selection of the patient cohort, but it could not be avoided in a STEMI network organization. Although patients with previous MI were not excluded, they were not significantly different in the LVR and non-LVR subgroups. Because our study did not include cardiac magnetic resonance, we were unable to determine the direct influence of microvascular obstruction and presence of myocardial hemorrhage on LVR. Myocardial strain evaluation was performed using echo scanners of a single vendor; because inter-vendor strain calibration is still lacking [31,32], study results may not necessarily apply to strain calculated using echo scanners from other vendors. The LVR was evaluated using 2D echocardiography, although its estimation using 3D echocardiography [33] and cardiac magnetic resonance [1] can be more accurate. However, 2D echocardiography is still the method more commonly used for LVR assessment in current clinical practice. The outcome endpoint was a secondary endpoint, and our retrospective analysis was based on a limited number of clinical events during the follow-up period; thus, we tried to avoid excessive statistical analysis and overinterpretation of data.

## 5. Conclusions

In patients treated for STEMI, acute %SAS is significantly and independently associated with development of LVR within 1 year and is better than other echocardiographic LVR predictors; acute %SAS is also associated with unfavorable clinical outcome. Thus, assessment of %SAS after primary PCI could be a helpful routine evaluation to identify very early those STEMI patients at higher risk. Prospective multicenter studies on a wider cohort of STEMI patients are needed to confirm our observations.

## Figures and Tables

**Figure 1 jcm-11-05348-f001:**
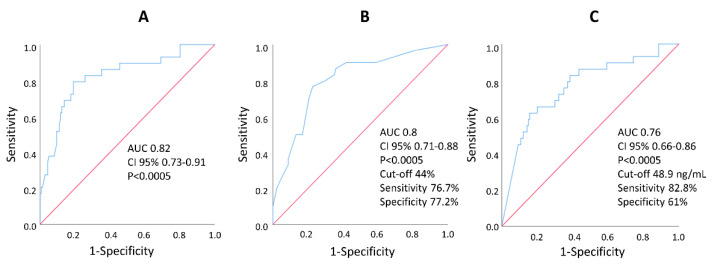
Receiver operating characteristic curves showing the capability of multivariable model 1 (**A**), baseline %SAS (**B**) and baseline peak troponin I (**C**) to discriminate patients with and without left ventricular remodeling. AUC, area under the curve; CI, confidence interval.

**Figure 2 jcm-11-05348-f002:**
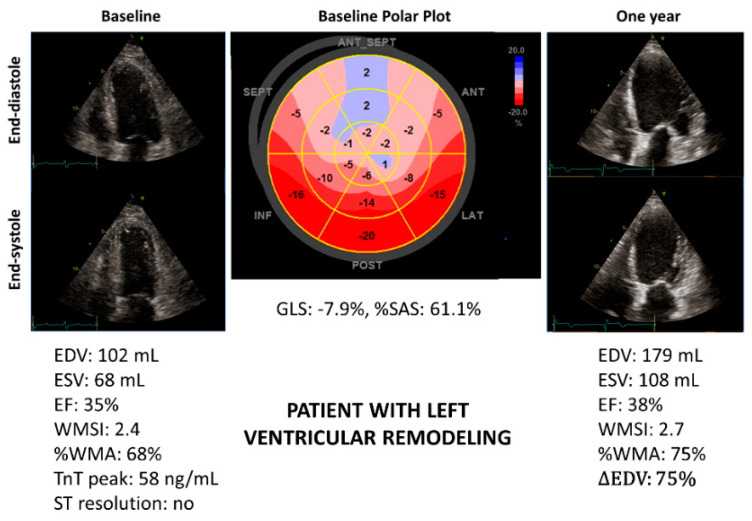
Patient with anterior ST-elevation myocardial infarction (STEMI) showing severe impairment of left ventricular systolic function and high %SAS at baseline. The polar plot in the middle shows the distribution of longitudinal myocardial strain throughout the LV, which is divided into six basal, mid-portion and apical segments. The numbers on each myocardial segment are the longitudinal myocardial strain values. After 12 months, left ventricular remodeling occurred. EDV, end-diastolic volume; ∆EDV: difference in EDV as a percentage of the basal value; EF, ejection fraction; ESV, end-systolic volume; GLS, global longitudinal strain; %SAS, percentage of severely altered strain; Tn, troponin; %WMA, percentage of the extent of wall motion abnormalities; WMSI, wall motion score index.

**Figure 3 jcm-11-05348-f003:**
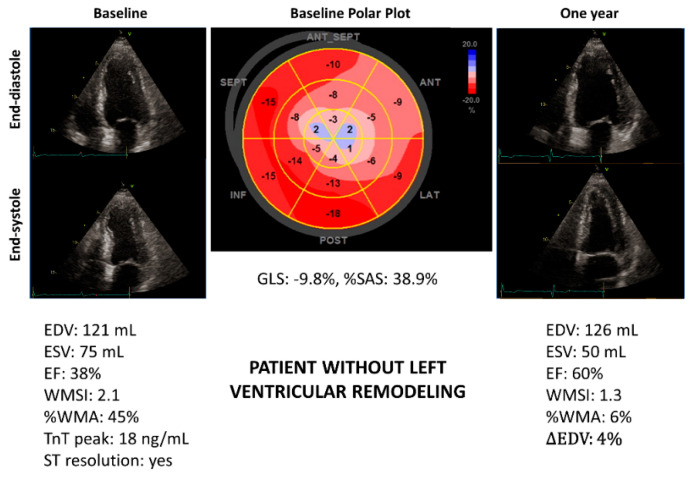
Patient with anterior ST-elevation myocardial infarction (STEMI) showing severe impairment of left ventricular systolic function but lower %SAS at baseline in comparison with patient of Figure 2. After 12 months, no left ventricular remodeling occurred. EDV, end-diastolic volume; ∆EDV: difference in EDV as a percentage of the basal value; EF, ejection fraction; ESV, end-systolic volume; GLS, global longitudinal strain; %SAS, percentage of severely altered strain; Tn, troponin; %WMA, percentage of the extent of wall motion abnormalities; WMSI, wall motion score index.

**Figure 4 jcm-11-05348-f004:**
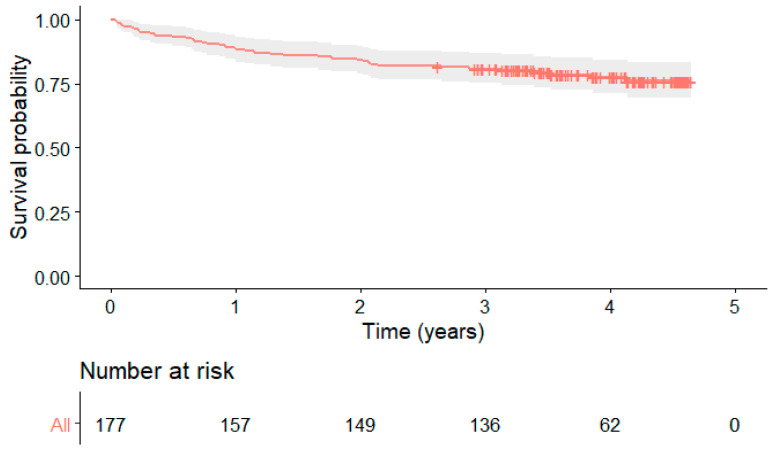
Survival probability free of clinical events (death from any cause and re-hospitalization for heart failure or acute coronary syndrome).

**Table 1 jcm-11-05348-t001:** Baseline characteristics of overall patients and subgroups with and without left ventricular remodeling.

	Overall (*n* = 177)	LVR Patients (*n* = 30, 16.9%)	Non-LVR Patients (*n* = 140, 79%)	*p* Value
Age (years)	67 (60–76)	69 (59–80)	66 (60–75)	0.668
Males (*n*)	128 (72.3%)	19 (63.3%)	105 (75%)	0.193
BSA (m^2^)	2 (1.8–2.1)	2 (1.7–2.1)	2 (1.8–2.1)	0.976
BMI (kg/m^2^)	26.5 (24.6–30.1)	27.1 (24.2–30.5)	26.5 (24.9–30)	0.932
Smoker (*n*)	101 (57.1%)	14 (46.7%)	83 (59.3%)	0.201
Diabetes (*n*)	36 (20.3%)	6 (20%)	24 (17.1%)	0.712
CV Family history (*n*)	50 (28.2%)	11 (36.7%)	38 (27.1%)	0.293
Hypertension (*n*)	111 (62.7%)	23 (76.7%)	82 (58.6%)	0.064
Dyslipidemia (*n*)	80 (45.2%)	13 (36.7%)	66 (47.1%)	0.654
COPD (*n*)	5 (2.8%)	0	3 (2.1%)	0.416
LDL (mg/dL)	126 (104–158)	124 (94–163)	126 (107–155)	0.946
Glycated Hb (mmol/mol)	40 (36.5–44)	41 (36–50)	39 (36–43)	0.194
GFR (mL/min)	89 (64–109)	92 (62–107)	88 (65–110)	0.928
Peak CK-MB (ng/mL)	159.6 (87.1–256)	243.9 (170–300)	145.6 (83.6–233)	0.001
Peak TnI (ng/mL)	39.2 (18.6–70.2)	76.5 (51.6–80)	35.8 (17.8–61.7)	<0.0005
Systolic blood pressure (mmHg)	130 (120–145)	123 (110–145)	130 (120–145)	0.15
Heart rate (bpm)	75 (62–80)	72 (58–80)	74 (62–80)	0.682
Killip class				
1–2 (*n*)	168 (94.9%)	27 (90.0%)	137 (97.9%)	0.034
3–4 (*n*)	9 (5.1%)	3 (10.0%)	3 (2.1%)	
Previous infarct	21 (12.3%)	5 (17.9%)	15 (11.0%)	0.302
Infarct site				
Anterior (*n*)	79 (44.9%)	17 (56.7%)	58 (41.7%)	0.051
Lateral (*n*)	36 (20.5%)	9 (30.0%)	26 (18.7%)	
Inferior (*n*)	54 (30.7%)	4 (13.3%)	48 (34.5%)	
Inferior and right ventricle (*n*)	7 (4.0%)	0	7 (5.0%)	
Culprit coronary vessel				
LAD (*n*)	77 (43.8%)	17 (56.7%)	56 (40.3%)	0.263
CCA (*n*)	32 (18.2%)	5 (16.7%)	26 (18.7%)	
RCA (*n*)	61 (34.7%)	7 (23.3%)	52 (37.4%)	
Left main (*n*)	4 (2.3%)	0 (0%)	4 (2.9%)	
IA (*n*)	2 (1.1%)	1 (3.3%)	1 (0.7%)	
Number of deceased coronary vessels				
One (*n*)	65 (36.7%)	12 (40.0%)	52 (37.1%)	0.256
Two (*n*)	61 (34.5%)	7 (23.3%)	53 (37.9%)	
Three (*n*)	51 (28.8%)	11 (36.7%)	35 (25.0%)	
Final TIMI flow				
0 (*n*)	1 (0.6%)	0	0	0.377
1 (*n*)	1 (0.6%)	0	1 (0.7%)	
2 (*n*)	5 (2.8%)	2 (6.7%)	3 (2.1%)	
3 (*n*)	170 (96.0%)	28 (93.3%)	136 (97.1%)	
Time from FMC to PCI (minutes)	57 (43–89)	63 (51–146)	54 (40–78)	0.051
Time from symptoms’ onset to diagnosis				
<3 h (*n*)	150 (84.7%)	22 (73.3%)	124 (88.6%)	0.036
≥3 h (*n*)	27 (15.3%)	8 (26.7%)	16 (11.4%)	
ECG ST-resolution (>50%) (*n*)	110 (62.1%)	12 (40%)	95 (67.9%)	0.004
Primary PCI (*n*)	177 (100%)	30 (100%)	140 (100%)	-
Staged PCI (*n*)	67 (37.9%)	9 (30%)	57 (40.7%)	0.272
Complete revascularization (*n*)	7 (4.0%)	1 (3.3%)	6 (4.3%)	0.814
Residual vessels after staged PCI (*n*)	1 (1–2)	1 (1–2)	1 (1–2)	0.584

BMI, body mass index; bpm, beat per minute; BSA, body surface area; CCA, circumflex coronary artery; CK-MB, creatine kinase-MB; COPD, chronic obstructive pulmonary disease; CV, cardiovascular; FMC, first medical contact; ECG, electrocardiogram; GFR, glomerular filtration rate; Hb, hemoglobin; IA, intermediate coronary artery; LAD, left anterior descending artery; LDL, low density lipoprotein; LVR, left ventricular remodeling; PCI, percutaneous coronary intervention; RCA, right coronary artery; TIMI, thrombolysis in myocardial infarction; Tn, troponin. For each variable, the median value with interquartile range is reported, unless differently indicated.

**Table 2 jcm-11-05348-t002:** Echocardiographic characteristics of overall patients at time T0, T1 and T2.

	T0	*n*	T1	*n*	T2	*n*	*p* Value
EDV (mL)	99 (86–116)	177	99 (85.5–117)	172	98 (85–120)	167	0.335
ESV (mL)	50 (42–65)	177	43 (34.5–55)	172	42 (33–56)	167	* <0.0005 † <0.0005 ‡ 0.465
EF (%)	49 (41–53)	177	56 (49–62)	172	58 (50–62)	167	* <0.0005 † <0.0005 ‡ 0.129
GLS (%)	−13.2 (−15.6–−10.3)	177	−16.9 (−18.6–−13.8)	170	−17.9 (−20.4–−15.5)	163	* <0.0005 † <0.0005 ‡ <0.0005
%SAS	29 (6–53)	173	0 (0–18)	165	0 (0)	163	* <0.0005 † <0.0005 ‡ 0.021
Left atrial volume index (mL/m^2^)	29 (24–35)	164	27 (21–34)	171	31 (25–36)	152	* <0.0005 † 0.038 ‡ 0.038
MR grade							
No MR (*n*)	57 (35.2%)	162	85 (53.5%)	159	79 (55.2%)	143	* <0.0005
Mild (*n*)	79 (48.8%)		61 (38.4%)		51 (35.7%)		
Moderate (*n*)	24 (14.8%)		13 (8.2%)		12 (8.4%)		
Severe (*n*)	2 (1.2%)		0		1 (0.7%)		
WMSI	1.6 ± 0.4	176	1.6 ± 0.3	166	1.3 ± 0.3	164	* <0.0005 † <0.631 ‡ <0.0005
%WMA	1.7 (1.4–2)	176	1.6 (1.4–1.9)	166	1.2 (1–1.6)	164	* <0.0005 † 0.063 ‡ <0.0005
E/A ratio	0.8 (0.6–1.1)	151	0.8 (0.7–1.1)	142	0.9 (0.7–1.1)	161	* 0.247
E/e’ ratio	9.2 (6.9–12.1)	70	8.4 (6.7–11.6)	63	8.1 (6.2–10.1)	157	* 0.003 † 0.264 ‡ 0.264

T0, baseline; T1, 3 months; T2, 12 months. EDV, end-diastolic volume; EF, ejection fraction; ESV, end-systolic volume; GLS, global longitudinal strain; MR, mitral regurgitation; %SAS, percentage of severely altered strain; %WMA, percentage of the extent of wall motion abnormalities; WMSI, wall motion score index. *p* *, comparison between T0, T1 and T2; *p* †, comparison between T0 and T1; *p* ‡, comparison between T1 and T2. For each variable, the median value with interquartile range is reported, unless differently indicated.

**Table 3 jcm-11-05348-t003:** Echocardiographic characteristics of patients with and without LVR at T0, T1 and T2.

	T0			T1			T2		
	LVR	Non-LVR	*p* Value	LVR	Non-LVR	*p* Value	LVR	Non-LVR	*p* Value
EDV (mL)	107 (85–126)	96 (86–116)	0.260	133 (105–159)	96 (84–112.5)	<0.0005	134 (114−175)	95 (83–107)	<0.0005 * <0.0005 † 0.148
ESV (mL)	60 (52–75)	47 (41–59)	<0.0005	74 (61–93)	41 (34–49)	<0.0005	77 (59–100)	38 (31–48)	<0.0005 * <0.0005 † <0.0005
EF (%)	41 (34–47)	49.8 (43–55)	<0.0005	42 (37–47)	58 (52–62)	<0.0005	42 (40–50)	59 (54–63)	<0.0005 * 0.005 † <0.0005
GLS (%)	−9.3 (−11.2 - −8.2)	−14.1 (−16.1–−11.6)	<0.0005	−12.1 (−13.1–−10.8)	−17.5 (−18.9–−15.2)	<0.0005	−13.5 (−15.9–−11.0)	−19 (−20.6– −17.0)	<0.0005 * <0.0005 † <0.0005
%SAS	56 (47–65)	18 (6–41)	<0.0005	35 (12–53)	0 (0–6)	<0.0005	12 (0–41)	0 (0)	0.001 * <0.0005 † <0.0005
Left atrial volume index (mL/m^2^)	31 (26–35)	29 (24–32)	0.222	30 (24–40)	27 (21–33)	0.061	36 (31–45)	30 (24–35)	<0.0005 * <0.0005 † 0.007
MR grade									
No MR (*n*)	7 (25.0%)	47 (37.0%)	0.432	8 (29.6%)	76 (58.5%)	0.023	9 (37.5%)	70 (58.8%)	0.046 * 0.325 † <0.0005
Mild (*n*)	15 (53.6%)	62 (48.8%)		16 (59.3%)	44 (33.8%)		11 (45.8%)	40 (33.6%)	
Moderate (*n*)	6 (21.4%)	16 (12.6%)		3 (11.1%)	10 (7.7%)		3 (12.5%)	9 (7.6%)	
Severe (*n*)	0 (0%)	2 (1.6%)		0 (0%)	0 (0%)		1 (4.2%)	0 (0%)	
WMSI	2 (1.6–2.2)	1.6 (1.4–1.9)	<0.0005	2 (1.9–2.1)	1.5 (1.2–1.8)	<0.0005	1.7 (1.5–2.2)	1.1 (1–1.4)	<0.0005 * 0.148 † <0.0005
%WMA	50 (31.2–56.2)	31.2 (12.5–37.5)	<0.0005	50 (37.5–50)	18.7 (0–37.5)	<0.0005	37.5 (25–50)	0 (0–12.5)	<0.0005 * 0.008 † <0.0005
E/A ratio	1 (0.6–1.4)	0.8 (0.7–1)	0.595	1 (0.7–1.1)	0.8 (0.7-.1)	0.486	0.9 (0.7–1.3)	0.8 (0.6–1)	0.177 *0.368 †0.494
E/e’ ratio	10 (7–12)	9.1 (6.8–12)	0.973	9.5 (6.8–11.9)	8.3 (6.7–11.4)	0.421	9.3 (7.6–11.4)	7.7 (6.1–10)	0.012 * 0.607 † 0.004

T0, baseline; T1, 3 months; T2, 12 months. EDV, end-diastolic volume; EF, ejection fraction; ESV, end-systolic volume; GLS, global longitudinal strain; LVR, left ventricular remodeling; MR, mitral regurgitation; %SAS, percentage of severely altered strain; %WMA, percentage of the extent of wall motion abnormalities; WMSI, wall motion score index. LVR patients: *n* = 30 (17.6%), non-LVR patients: *n* = 140 (82.3%). * *p*, comparison between T0, T1 and T2 in LVR patients; † *p*, comparison between T0, T1 and T2 in non-LVR patients. For each variable, the median value with interquartile range is reported, unless differently indicated.

**Table 4 jcm-11-05348-t004:** Multivariable logistic regression analysis for left ventricular remodeling.

	Model 1 Including %SAS			Model 2 Including WMSI			Model 3 Including %WMA		
	OR	95% CI	*p* Value	OR	95% CI	*p* Value	OR	95% CI	*p* Value
Time from symptoms’ onset to diagnosis ≥3 h	-	-	0.194	-	-	0.501	-	-	0.501
Peak TnI (ng/mL)	1.025	1.004–1.047	0.022	1.030	1.009–1.051	0.004	1.030	1.009–1.051	0.004
MRA at discharge	-	-	0.821	-	-	0.969	-	-	0.969
ECG ST-resolution >50%	-	-	0.537	-	-	0.894	-	-	0.894
EF (%)	-	-	0.051	-	-	0.136	-	-	0.136
ESV (mL)	-	-	0.642	-	-	0.775	-	-	0.775
%SAS (for 5% increase)	1.226	1.098–1.369	<0.0005	-	-	-	-	-	-
GLS (%)	-	-	0.057	1.413	1.186–1.683	<0.0005	1.413	1.186–1.683	<0.0005
WMSI	-	-	-	-	-	0.310	-	-	-
%WMA	-	-	-	-	-	-	-	-	0.060

CI, confidence interval; ECG, electrocardiogram; EF, ejection fraction; ESV, end-systolic volume; GLS, global longitudinal strain; MRA, mineralocorticoid receptor antagonist; OR, odds ratio; %SAS, percentage of severely altered strain; Tn, troponin; %WMA, percentage of the extent of wall motion abnormalities; WMSI, wall motion score index.

## Data Availability

The data of the study are available upon justified request.

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
