# Peer review of "Severe Impairment of Left Ventricular Regional Strain in STEMI Patients Is Associated with Post-Infarct Remodeling"

_jcm, 2022, doi:10.3390/jcm11185348_

Round 1

Reviewer 1 Report

Luisi et al investigated the association between %SAS and LVR in patients with STEMI. In addition, they also assessed the association between %SAS and clinical events. They found that %SAS is associated with post-infarct LVR and could predict unfavorable clinical outcomes. It is well written. And is an interesting study. I have some comments that should be addressed by the authors.

1.     1.  Were the individuals who acquired and/or analyzed the data blinded to the clinical picture of the individuals?

2.    2.  It is not clear how many patients were selected to do the reproducibility analysis for %SAS calculation. How to do the intra-observer and inter-observer variability. I also only see one statistical analysis approach with respects to reproducibility of %SAS (ICC). It's hard to justify the results based on just one statistical approach.

3.     3. Survival curves including %SAS should be obtained in a Kaplan-Meier analysis and compared using the log-rank test.

4.     4.  I did not see which variables were included to multivariate Cox analysis. To demonstrate a superior prognostic value of %SAS, I would suggest the Authors to build different multivariable prognostic models, each for %SAS, GLS, WMSI and %WMA, and compare their C-index and Akaike information criterion (AIC).

Author Response

Reviewer # 1

Luisi et al investigated the association between %SAS and LVR in patients with STEMI. In addition, they also assessed the association between %SAS and clinical events. They found that %SAS is associated with post-infarct LVR and could predict unfavorable clinical outcomes. It is well written. And is an interesting study.

We thank this Reviewer for his/her favorable comments.

I have some comments that should be addressed by the authors.

  1. Were the individuals who acquired and/or analyzed the data blinded to the clinical picture of the individuals?

The investigators who analyzed the images were blinded to the clinical patients’ information. This is now stated in the text at p. 3. Image acquisition at T0 was performed in the intensive care unit at patient bedside, thus individuals who acquired the images could not be blinded to the clinical picture of the patients.

  1. It is not clear how many patients were selected to do the reproducibility analysis for %SAS calculation. How to do the intra-observer and inter-observer variability. I also only see one statistical analysis approach with respects to reproducibility of %SAS (ICC). It's hard to justify the results based on just one statistical approach.

Observer variability analysis was performed in 20 patients randomly selected from the entire cohort at T0 using the intraclass correlation coefficient and the Bland-Altman analysis. Results are reported in paragraph 3.5.

  1. Survival curves including %SAS should be obtained in a Kaplan-Meier analysis and compared using the log-rank test. 

The log-rank test only compares survival between groups, while in the present work the %SAS was analyzed as continuous variable. This is the reason why Cox Proportional Hazard model was run to evaluate the role of %SAS on the composite endpoint. Furthermore, it provides a measure of the magnitude of the association between %SAS and the endpoint of interest, i.e., the Hazard Ratio, which is an added value compared to the log rank test which only tells if there is a survival difference between the groups of interest.

  1. I did not see which variables were included to multivariate Cox analysis.

Variables for multivariate Cox analysis for the secondary objective were LVR, %SAS and peak of troponin. This was reported in paragraph 3.4. 

To demonstrate a superior prognostic value of %SAS, I would suggest the Authors to build different multivariable prognostic models, each for %SAS, GLS, WMSI and %WMA, and compare their C-index and Akaike information criterion (AIC).

The outcome endpoint was a secondary endpoint and our retrospective analysis was based on a limited number of clinical events during the follow-up period, thus we tried to avoid excessive statistical analysis and overinterpretation of data. This is now stated in paragraph 4.5.

Reviewer 2 Report

Luisi et al. investigated the percentage of severely altered strain (%SAS) as a predictor for left ventricular remodeling (LVR) and the clinical outcome. They found that increased %SAS is associated with increased LVR and poorer clinical outcome. The study is relevant to the community. The manuscript is well-written. The study is descriptive and has different limitations that were already discussed by the authors in an appropriate way.

The authors should discuss/compare %SAS to other predictors for LVR and poor clinical outcome in more detail.

Author Response

Reviewer # 2

Luisi et al. investigated the percentage of severely altered strain (%SAS) as a predictor for left ventricular remodeling (LVR) and the clinical outcome. They found that increased %SAS is associated with increased LVR and poorer clinical outcome. The study is relevant to the community. The manuscript is well-written. The study is descriptive and has different limitations that were already discussed by the authors in an appropriate way.

We thank this Reviewer for his/her favorable comments.

The authors should discuss/compare %SAS to other predictors for LVR and poor clinical outcome in more detail.

We added the following text (paragraph 4.2) to clarify the role of other predictors: “Various echocardiographic measures of LV global systolic function, including LV-EF, ESV, WMSI and GLS, have been proposed over the years to predict, during the early phase of STEMI, subsequent development of LVR or outcome (3-6,15,16). Results, however, were not univocal. For example, in a meta-analysis 2D GLS, evaluated within 48 hours after the acute event, has been shown to predict adverse LVR after STEMI (defined as increase in LV-EDV and/or LV-ESV by 15 to 20%) (6). Other authors, however, found that, in high-risk STEMI patients, GLS was not superior to conventional echocardiography in predicting outcome (17).

Round 2

Reviewer 1 Report

I have no further comment.